# Beyond Text: LLM-Based Multimodal and Cross-Lingual Transfer Learning Framework for Low-Resource Tigrigna Sentiment Analysis

## Abstract

Sentiment analysis in low-resource languages remains underexplored, particularly for Tigrigna, where communication frequently combines text, emojis, and memes. We introduce TigXMM, a cross-lingual, multilingual, and multimodal framework for Tigrigna sentiment analysis, along with the first multimodal sentiment dataset for this language. The dataset, collected from social media, integrates text, emojis, and meme content to capture real-world communication patterns. We benchmark widely used multilingual models, including mBERT, AfriBERTa, XLM-RoBERTa, XLNet, BLOOMZ, and LLaMA, and highlight their limitations in processing multimodal signals. To address these challenges, we design an LLM-based cross-lingual transfer model with multimodal adapters for text, emoji, and meme fusion using hybrid attention and additive–hierarchical strategies. Experimental results demonstrate that our approach consistently improves sentiment classification performance: from 78.4% accuracy on text-only inputs to 81.2% with emojis, 86.3% with memes, and 89.7% when combining all modalities, achieving state-of-the-art performance for Tigrigna sentiment analysis. Beyond performance gains, this work contributes the first multimodal dataset and a reproducible framework, providing open resources to advance sentiment analysis for underrepresented African languages.

## 1 Introduction

Modern online communication has become more multimodal, encompassing not only written text but also visual and symbolic elements such as emojis, images, GIFs, memes, and videos. These elements are used across diverse platforms, including messaging apps, social media, blogs, and digital forums, to convey sentiment, emotion, and intent in a way that often transcends plain text (Zhou et al., 2018; Felbo et al., 2017). Memes or emojis can enhance or alter the emotional tone of a message, while memes often condense complex cultural commentary into a single, shareable image (Barbieri et al., 2018; Milosavljević, 2020; Diengdoh, 2024). This fusion of modalities introduces significant challenges for sentiment analysis, as traditional models are typically optimized for text-only input and fail to capture the full spectrum of affective cues used in contemporary communication amplified in low-resource languages (Choudhury & Deshpande, 2021).

The majority of sentiment research has focused on high-resource languages such as English, Chinese, and Spanish, where large annotated corpora and pretrained models are readily available (Pakray et al., 2025). Low-resource languages, including those in Africa such as Tigrigna, Amharic, Hausa, Yoruba, and Swahili, remain significantly underexplored. Tigrigna, also written as Tigrigna, is a Semitic language spoken by over 10 million people across Eritrea and Ethiopia (Fesseha et al., 2021; Tela et al., 2024). The other crucial limitation of existing African sentiment research is its almost exclusive reliance on text-only corpora. This imbalance has created a digital language divide, limiting the accessibility of NLP technologies to speakers of underrepresented languages. current initiatives like AfriSenti-SemEval (Muhammad et al., 2023a;b) have started to close this gap by releasing

benchmark sentiment datasets for multiple African languages. However, Tigrigna remains critically under-resourced and lacks multimodal resources that reflect real-world communication practices.

Cross-lingual transfer has emerged as a key strategy for addressing low-resource limitations. mBERT (Devlin et al., 2019) was the first widely used multilingual transformer, but its performance degrades for underrepresented languages. AfriBERTa (Ogueji, 2022), trained specifically on African languages, improves coverage but remains primarily text-focused. BLOOMZ (Muennighoff et al., 2023), an instruction-tuned multilingual LLM, demonstrates stronger zero-shot generalization but lacks explicit adaptation to African contexts. XLM-RoBERTa (Conneau et al., 2020)has shown strong cross-lingual representation learning, yet requires fine-tuning for optimal performance. Finally, XLNet (Tela, 2020), based on permutation language modeling, provides diversity in baselines but has limited adaptation to multilingual settings (Tela et al., 2024). While these models offer valuable cross-lingual insights, they are generally designed for text-only transfer, with little consideration of multimodal signals.

The emergence of Large Language Models has shifted the landscape of low-resource NLP (Zhao et al., 2024; Gebremeskel & Feng, 2025). LLaMA (Touvron et al., 2023; Dubey et al., 2024), in particular, is noted for its efficiency, multilingual adaptability, and suitability for fine-tuning in resource-constrained environments. Recent efforts have adapted LLMs for low-resource languages through fine-tuning and prompt-based learning (Zhao et al., 2024). However, direct fine-tuning often fails without architectural adjustments (such as vocabulary changes) and sufficient in-language data. Our work addresses this by combining vocabulary expansion with efficient fine-tuning (Dettmers et al., 2023) to tailor LLaMA for Tigrigna. Full fine-tuning of LLMs is computationally prohibitive. PEFT methods, such as SentencePiece Adapters (Kudo & Richardson, 2018), Prefix-tuning (Woldeyohannis & Meshesha, 2022), and parameter-efficient fine-tuning, notably LoRA (Hu et al., 2022), which injects trainable low-rank matrices into attention layers, have become the standard for adaptation. The injection of attention and multilayer perceptron projections to reduce trainable parameters without large accuracy is widely used in low-resource fine-tuning (Hu et al., 2023). These approaches established efficient, competitive open LLMs trained on web-scale corpora, which have made LLaMA highly practical for domain-specific and low-resource tasks (Hu et al., 2023; Weyssow et al., 2025). These techniques enable targeted adaptation without requiring prohibitively expensive full-parameter fine-tuning, making them attractive for languages like Tigrigna. Their application in a multi-stage, cross-lingual context for a specific language family is a key area of our exploration. However, applications of LLaMA to multimodal and culturally specific sentiment tasks remain underexplored. Emojis carry systematic sentiment polarity and semantics; classic resources provide polarity rankings and show temporal contextual variation and embedding methods (Reelfs et al., 2020). Meme understanding is harder: ironic visuals, culture-specific references, and OCR noise complicate sentiment. Foundational multimodal V&L models (CLIP, VisualBERT, ViLBERT) align images and text through contrastive or joint Transformer encoders, offering blueprints for meme/text fusion (Hu et al., 2022; Runwal et al., 2025). Benchmark datasets such as Hateful Memes and Memotion standardized evaluation of vision-language models for harmful content and meme affect, but concentrate on English and high-resource contexts, leaving African languages underrepresented (Hu et al., 2022; Shi & Lipani, 2023).

To our best knowledge, there is no prior work that has released a Tigrigna meme and/or emoji-aware sentiment dataset or evaluated multimodal fusion with LLM-based PEFT for cross-lingual transfer; this is the gap our study addressed. The Cross-lingual transfer learning from high-resource languages offers a promising path for low-resource sentiment analysis. Multilingual models such as mBERT, XLM-RoBERTa, AfriBERTa, and BLOOMZ have shown success in leveraging high-resource language supervision for African languages. However, these models often struggle with Tigrigna due to its morphological complexity, orthographic variation, and code-switching tendencies, compounded by the scarcity of annotated data. Recently, large language models like LLaMA and LLaMA-2 have demonstrated remarkable adaptability in multilingual transfer and parameter-efficient fine-tuning. Techniques such as LoRA and other PEFT approaches make it feasible to adapt these models to low-resource settings with limited compute, opening opportunities for multimodal sentiment analysis in underrepresented languages. To address these challenges, we introduce TigXMM,

a cross-lingual, multilingual, and multimodal framework equipped with adapters for text, emoji, and meme representations, combined through hybrid fusion strategies. Alongside the model, we construct the first multimodal sentiment dataset for Tigrigna, collected from social media and annotated across text, emoji, and meme modalities. Our contributions establish both the resources and methodological foundations for advancing sentiment analysis in underrepresented languages. This study has been guided through the following four progressive sequence from language transfer (RQ1), modality integration (RQ2 & RQ3), and generalization to other low-resource languages (RQ4) research questions:

- RQ1. How can cross-lingual transfer learning with large language models be adapted to improve sentiment analysis in low-resource languages such as Tigrigna?

- RQ2. To what extent do multimodal models enhance sentiment classification of recent online communication compared to text-only approaches?

- RQ3. How can a unified multimodal framework be developed to address the limitations of existing multilingual models and provide a benchmark for Tigrigna sentiment analysis?

- RQ4. Can multimodal approaches achieve state-of-the-art performance for underrepresented African languages?

Our work made several key contributions to advancing sentiment analysis in low-resource settings. First, we introduce the initial multimodal sentiment dataset for Tigrigna, which integrates text, emojis, and memes, thereby reflecting the authentic communication patterns of Tigrigna speakers on social media platforms. Second, we provide a comprehensive benchmarking study by evaluating widely used multilingual baselines, namely mBERT, AfriBERTa, XNet, BLOOMZ, and XLM-RoBERTa, on Tigrigna sentiment classification, establishing a strong performance reference for future research. Third, we propose TigXMM, a novel LLM-based cross-lingual transfer model enhanced with multimodal adapters, designed specifically to capture sentiment signals from text with emojis and memes in low-resource contexts. Fourth, through extensive experiments and ablation studies, we demonstrate that TigXMM achieves state-of-the-art performance, outperforming multilingual baselines while also offering insights into the challenges of multimodal sentiment analysis in underrepresented languages. Finally, we benchmark a detailed dataset creation and preprocessing pipeline for Tigrigna sentiment tasks, which we make available to support further community-driven research in African language technologies.

The rest of the paper is organized as follows: Section describes the proposed methodology, including dataset construction, emoji and meme integration, and the design of the proposed model. Section 3 presents experimental setup, baselines, and proposed model results, followed by Section 4, which discusses findings, challenges, and insights. Finally, the 5th Section concludes the paper and outlines promising directions for future research.

## 2 The TigXMM Proposed Model Framework

We propose TigXMM, a Tigrigna Cross-Lingual Multilingual and Multimodal with vision adapter Sentiment Analysis, a model designed to handle cross-lingual transfer from English and Amharic to Tigrigna and multimodal sentiment classification of text, emojis, and memes. The model incorporates the adapted vocabulary extension LLaMA, for cross-lingual learning, allowing Tigrigna to benefit from existing high-resource languages while also processing multimodal data (text with emoji, and/or image) to improve sentiment analysis. The proposed framework, illustrated in Figure 1, integrates cross-lingual adaptation and multimodal alignment through four components.

### 2.1 TigXMM Model Architecture

The proposed model builds upon the LLaMA backbone and extends it with cross-lingual transfer capabilities, Masked Language Modeling on a mixture of Tigrigna and English texts enhanced with a contrastive loss, and a vision adapter. Parameter-efficient fine-tuning methods LoRA (Woldeyohannis & Meshesha, 2022) and QLoRA (Dettmers et al., 2023) were

employed to adapt the model to Tigrigna under resource constraints. Cross-lingual transfer was achieved by leveraging knowledge from English and Amharic to improve performance on Tigrigna data. Two specialized components were incorporated to handle non-textual sentiment cues. First, an emoji-aware embedding layer was introduced to integrate Unicode emoji polarity scores into the token stream, allowing the model to interpret sentiment-rich emoji sequences. Second, a meme-handling module was developed, which fused OCR-extracted text, captions, hashtags, and visual features from images into the LLaMA encoder using a multimodal fusion adapter.

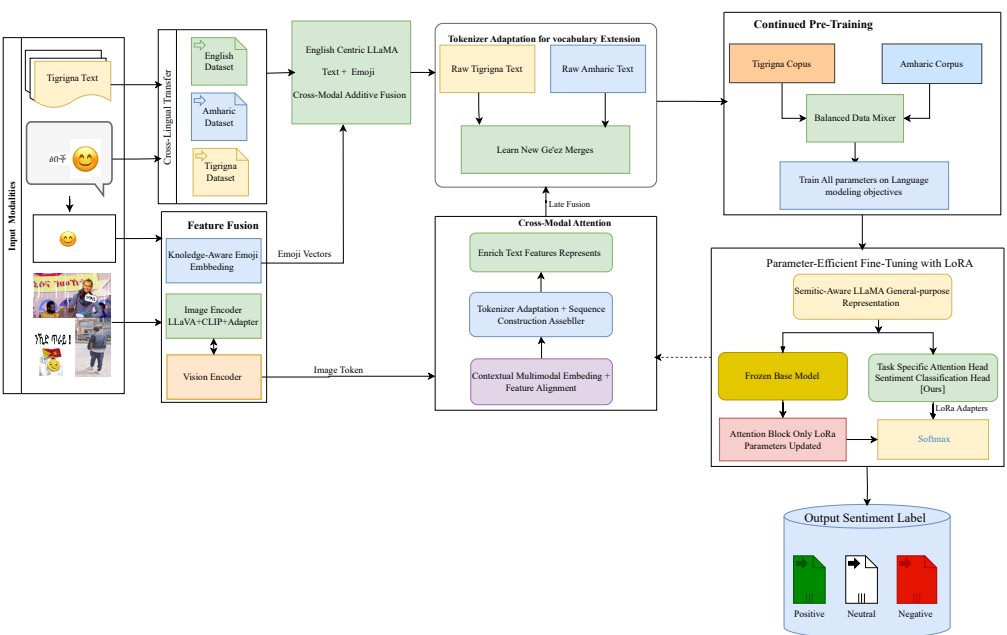

Figure 1: The TigXMM Framework Architecture. A LLaMA backbone enabled Cross-lingual transfer and multimodal input features aligned via Hybrid additive and cross-attention fusion adapters with parameter-efficient LoRA for Tigrigna sentiment analysis.

## 2.2 Training and Cross-Lingual Transfer Strategy

The entire model (except for the pre-trained encoders, which are fine-tuned) is trained end-to-end on a small, manually annotated dataset of Tigrigna social media. The cross-lingual transfer occurs primarily through the multilingual LLM (Text Encoder). Its pre-trained weights provide a strong prior for understanding Tigrigna by relating it to other languages it was trained on, dramatically reducing the amount of labeled Tigrigna data needed for effective performance. The other encoders and the fusion modules learn to align their features with this powerful textual representation. Our training follows a three-stage strategy:

- Stage 1: Pretraining on large multilingual corpora in English and Amharic, leveraging related high-resource languages for transfer.
- Stage 2: LoRA-based fine-tuning on the curated multimodal Tigrigna dataset, ensuring adaptation to domain-specific and script-specific features.
- Stage 3: Multimodal alignment, where text, emoji, and vision features are integrated through hierarchical fusion and jointly optimized for sentiment prediction.

## 2.3 Evaluation Metrics

Evaluation was conducted across text-only, emoji, and multimodal settings. For text-only experiments, we used standard-based Accuracy, Precision, Recall, and F1-score, with Macro-

F1 used to account for class imbalance, especially in zero- and few-shot experiments, and robustness-oriented metrics suited for multimodal, low-resource sentiment analysis. The Micro-F1 aggregates counts across classes before computing F1, which emphasizes performance on high-frequency classes.

$$\text{Micro-F1} = \frac{2 \cdot \sum_c TP_c}{2 \cdot \sum_c TP_c + \sum_c FP_c + \sum_c FN_c} \tag{1}$$

where $TP_c$, $FP_c$, and $FN_c$ symbolize the true positives, false positives, and false negatives for class $c$. The multimodal robustness-oriented metrics for the emoji-rich dataset, we introduced *Emoji Sentiment Coverage (ESC)*, which measures the model's ability to interpret emoji-only or emoji-dense posts correctly. For memes and image-text fusion, we evaluated *Meme Sentiment Consistency (MSC)*, defined as the degree of agreement between the model's predictions and human annotations across both textual and visual layers.

Formally, let $M$ stand for the set of all modality combinations (Text, Emoji, Meme, Hybrid), and $E$ be the set of samples containing at least one emoji. Each emoji $e \in E$ is assigned a prior sentiment polarity $s(e)$ from an emoji sentiment lexicon (derived from EmoLex, mapped into Tigrigna). For sample $i$, let $\hat{y}_i$ be the predicted sentiment.

$$\text{ESC} = \frac{1}{|\mathcal{E}|} \sum_{i \in \mathcal{E}} \mathbf{1}[\hat{y}_i = s(e_i)] \tag{2}$$

where $e_i$ is the dominant emoji in sample $i$, $\hat{y}_i$ is the model's predicted sentiment, and $s(e_i)$ is the prior sentiment polarity of $e_i$ from the emoji sentiment lexicon.

In addition, let $\hat{y}_i^{\text{fusion}}$ be the prediction for meme $i$ using the fused embedding, and let $\hat{y}_i^{\text{text}}$, $\hat{y}_i^{\text{image}}$, and $\hat{y}_i^{\text{ocr}}$ be the predictions from the unimodal channels.

$$\text{MSC} = \frac{1}{|\mathcal{M}|} \sum_{i \in \mathcal{M}} \mathbf{1}\Big[\hat{y}_i^{\text{fusion}} = \text{majority}\Big(\hat{y}_i^{\text{text}}, \hat{y}_i^{\text{image}}, \hat{y}_i^{\text{ocr}}\Big)\Big] \tag{3}$$

where $\mathcal{M}$ denotes the set of meme samples. This multi-faceted evaluation provides a richer and fairer assessment compared to unimodal or purely text-based baselines. A higher robustness score indicates greater stability under modality shifts, which is crucial in noisy, real-world Tigrigna social media contexts.

## 3 EXPERIMENT SETUP

This section details the experimental setup, dataset, baseline models, ablation studies, and evaluation metrics used to assess our proposed multimodal sentiment-aware framework for Tigrigna.

### 3.1 DATASET

We curated several complementary datasets to capture the diverse ways in which sentiment is expressed in Tigrigna social media. The text-only dataset consists of 68,596 samples collected from Facebook pages, Twitter posts, Telegram groups, and YouTube user comments. This corpus provides an enhancement to classical sentiment classification.

The emoji-only 160 samples dataset curated consists of posts and comments composed entirely of emojis and was injected with knowledge-awareness. A larger text + emoji dataset of approximately 5,000 manually and semi-supervised XLM-R annotated 10,000 samples was also constructed, capturing sentiment expressed through a mixture of Ge'ez-script text and emojis. Beyond text and emojis, a dedicated meme dataset of 2,000 samples was curated, and we built a text + memes dataset of 5,000 samples, which contains OCR-extracted texts, captions, hashtags, emojis, and visual features annotated by five native speakers with Cohen's k=0.80 inter-annotator agreement, providing the first balanced multimodal benchmark for sentiment analysis in a Ge'ez-script low-resource language. The dataset split used had been 75% for training, 10% for validation, and 15% for testing. Together, these multimodal datasets enable the study of sentiment analysis beyond text-only scenarios.

Table 1: The Tigrigna Dataset Composition, Statistics and Contributions

| Dataset | Size | Novel Contribution | Key Features |
|---|---|---|---|
| Text-only | 68,596 | 55,765 Publicly available (prior work), and 12,831 augmented for this work | Facebook, X(Twitter), Telegram, YouTube comments |
| Emoji-only (knowledge-aware) | 160 | New (this work) | Posts composed entirely of emojis |
| Text + Emoji | 10,000 | New (this work) | Mixed Ge'ez-script + emoji posts |
| Text + Memes | 5,000 | New (this work) | Multimodal posts with images |
| Dedicated Meme | 2,000 | New (this work) | OCR text + visual features + metadata |

## 3.2 BASELINES

The baselines provide insight into how far text-only multilingual models can perform before integrating multimodal or sentiment-aware enhancements.

### 3.2.1 TEXT-ONLY BASELINES

To evaluate the effectiveness of our approach, we benchmarked against five widely used multilingual models. mBERT Devlin et al. (2019) served as the general multilingual baseline, while AfriBERTa Ogueji (2022) provided a more regionally tailored benchmark for African languages. BLOOMZ Muennighoff et al. (2023), an instruction-tuned multilingual large language model, was included to test performance under zero- and few-shot settings. XLM-RoBERTa Conneau et al. (2020), a high-performing multilingual variant of RoBERTa, was used, given its strong cross-lingual representations. Finally, XLNet Yang et al. (2019) was considered for diversity, although it is less optimized for low-resource language scenarios. The experiment result is illustrated in Table 2.

Table 2: Baseline performance on the Tigrigna text-only sentiment dataset.

| Model | Acc | Prec | Rec | F1-Score | Macro-F1 |
|---|---|---|---|---|---|
| mBERT Devlin et al. (2019) | 68.2 | 66.5 | 67.0 | 66.7 | 65.9 |
| AfriBERTa Ogueji (2022) | 71.4 | 70.2 | 71.0 | 70.6 | 70.1 |
| BLOOMZ Muennighoff et al. (2023) | 72.5 | 71.9 | 72.2 | 72.0 | 71.8 |
| XLM-RoBERTa Conneau et al. (2020) | 75.9 | 75.0 | 75.5 | 75.2 | 75.1 |
| XLNet Yang et al. (2019) | 63.8 | 62.7 | 63.0 | 62.8 | 62.2 |
| LLaMA 2-7B Touvron et al. (2023) | 73.4 | 74.1 | 73.6 | 73.2 | 72.8 |
| LLaMA 2-13B Touvron et al. (2023) | 75.8 | 76.5 | 76.0 | 75.5 | 75.1 |
| LLaMA 3-7B Dubey et al. (2024) | 77.0 | 77.6 | 77.1 | 76.8 | 76.4 |
| LLaMA 3-13B Dubey et al. (2024) | 77.6 | 78.5 | 77.9 | 77.4 | 77.0 |
| **TigLLaMA (Ours)** | **89.0** | **89.7** | **89.1** | **89.3** | **89.0** |

Note: Acc = Accuracy, Prec = Precision, Rec = Recall, and the dataset used for this experiment is multi-modal.

### 3.2.2 MULTIMODAL BASELINES

To assess the effectiveness of existing cross-modal models, we compare against CLIP (contrastive vision–language pretraining) Radford et al. (2021), LLaVA (large language and vision alignment) Liu et al. (2023), and Flamingo (few-shot multimodal reasoning) Alayrac et al. (2022) as in Table 3. The results highlight the comparative performance of existing multimodal models against our proposed framework. Zero-shot CLIP and few-shot LLaVA achieve moderate accuracy (61–65%), reflecting limited transferability to Tigrigna sentiment tasks. Flamingo shows slight gains (67% accuracy, 63% macro-F1), but still struggles

with cross-lingual adaptation. In contrast, TigXMM (ours) achieves 74% accuracy and 71% macro-F1, outperforming all baselines by a significant margin, confirming the effectiveness of integrating text, emoji, and meme signals for robust sentiment analysis in low-resource Tigrigna.

Table 3: Performance of multimodal models Fine-Tuning on the Tigrigna sentiment dataset.

| Model | Modalities | Setting | Accuracy | Macro-F1 |
|---|---|---|---|---|
| CLIP Radford et al. (2021) | meme + Text | Zero-shot | 61% | 57% |
| LLaVA Liu et al. (2023) | meme + Text | Few-shot | 65% | 61% |
| Flamingo Alayrac et al. (2022) | Multimodal | Few-shot | 67% | 63% |
| **TigXMM(Ours)** | Text + Emoji + meme | Few-shot | **74%** | **71%** |

### 3.3 Cross-Lingual Transfer Settings

To address the shortage of annotated sentiment data for Tigrigna, we have designed three complementary cross-lingual transfer setups. Each setup utilizes pretrained multilingual or related-language resources to improve model adaptation. We systematically evaluate transfer performance across three scenarios:

First, we implemented an English-to-Tigrigna transfer strategy. This approach uses English as the dominant source language for large pretrained language models and multimodal encoders such as CLIP, LLaVA, and Flamingo. The transfer relies on zero-shot or few-shot cross-lingual alignment, projecting Tigrigna inputs into embedding spaces predominantly shaped by English semantics. While English provides access to a wealth of sentiment datasets, we anticipated challenges due to semantic drift, stemming from the linguistic distance between English (an Indo-European language) and Tigrigna (an Afroasiatic Semitic language).

The second strategy, Amharic-to-Tigrigna transfer, capitalizes on the close linguistic and cultural ties between these two Semitic languages. Both languages share the Ge'ez script and exhibit significant lexical and morphological similarities. This proximity suggests that transferring knowledge from Amharic may achieve better alignment than English, particularly for culturally specific sentiment expressions, idioms, and emoji usage patterns. By leveraging these linguistic and cultural parallels, the model can adapt more effectively to Tigrigna-specific nuances.

Finally, we explored a combined multilingual transfer approach that incorporates both English and Amharic, along with potentially other multilingual corpora such as XLM-R pretraining. This strategy aims to balance the rich multimodal coverage available in English with the script and cultural similarities offered by Amharic. We selected Amharic, a more researched Ethio-Semitic language, as a bridge language based on strong linguistic proximity to Tigrinya that shares Ge'ez script-based orthography, exhibits similar rich morphology, and has substantial lexical overlap and bilingual contact among speakers. By fine-tuning on this combined representation and subsequently applying Low-Rank Adaptation (LoRA) for Tigrigna, we hypothesize that the model can maximize performance. This approach investigates whether multilingual synergy, training on a mixture of high-resource languages before adapting to Tigrigna, can enhance low-resource sentiment transfer.

## 4 Results and Analysis

We analyze the performance of various models evaluated on the Tigrigna sentiment analysis task. Our analysis focuses on the effect of incorporating multimodal and cross-lingual learning strategies. The results demonstrate the effectiveness of leveraging multimodal inputs and LLM-based cross-lingual transfer in low-resource settings. The inclusion of emojis and memes led to a substantial performance boost through cross-lingual transfer, as Qualitative analysis revealed.

Table 4: The Performance of TigXMM Across Modalities

| Setting | Acc | Prec | Rec | F1 | Macro-F1 | ESC / MSC |
|---|---|---|---|---|---|---|
| Text-only | 78.4 | 77.5 | 77.9 | 77.7 | 77.6 | – |
| Text + Emojis | 81.2 | 80.5 | 80.7 | 80.6 | 80.4 | ESC = 84.1 |
| Text + Memes (OCR+Capt.) | 86.3 | 85.7 | 85.9 | 85.8 | 85.6 | MSC = 87.2 |
| Text + Emojis + Memes | 89.7 | 89.1 | 89.3 | 89.2 | 89.0 | MSC = 90.5 |

Note: Capt. = Caption; ESC = Emoji-Specific Context; MSC = Meme-Specific Context.

## 4.1 QUANTITATIVE FINDINGS

We next present results from our model. We conducted a systematic quantitative evaluation of our models across multiple baselines and ablations. To evaluate the effectiveness of TigXMM, we conducted comparative experiments against strong multilingual and cross-lingual baselines, including mBERT, AfriBERTa, XLM-RoBERTa, XLNet, BLOOMZ, and LLaMA. These models represent the current state of multilingual sentiment analysis and serve as appropriate benchmarks for assessing improvements in low-resource Tigrigna. The results highlight the performance gains achieved by incorporating multimodal signals and cross-lingual transfer, particularly when English pretraining is included. Comparative experiments show that while text-only models capture general sentiment cues, the integration of emoji and visual features provides measurable improvements in accuracy and F1-scores. The quantitative results Table 4 also reveal consistent challenges in handling sarcasm, idiomatic expressions, and ambiguous emoji usage, motivating the complementary error analysis presented above.

Table 5: Ablation Studies of TigXMM Modalities Performance

| Setting | Accuracy (%) | Macro-F1 (%) |
|---|---|---|
| Text-only | 78.4 | 77.6 |
| Text + Emojis | 81.2 | 80.4 |
| Text + Memes | 86.3 | 85.6 |
| Early Fusion | 84.7 | 83.9 |
| Late Fusion | 86.3 | 85.6 |
| LoRA Fine-tuning | 86.3 | 85.6 |
| Full Fine-tuning | 87.0 | 86.1 |
| Degraded OCR (20% error rate) | 80.5 | 79.7 |
| High-quality OCR (near-perfect) | 86.3 | 85.6 |

Note: The dataset used were multimodal text with emoji(s), text with meme(s), or text with meme and emoji.

## 4.2 ABLATION STUDIES

We conducted ablation studies to understand the contribution of each component: (i)Text-only vs Text+Emojis vs Text+Memes, (ii). Early fusion vs Late fusion, and (iii) Effect of OCR quality on sentiment classification. TigXMM achieves the best overall performance (86.8 F1), demonstrating the effectiveness of our complete architecture for low-resource Tigrinya sentiment analysis. We conducted ablation studies to understand the contribution of each components as shown in Table 5:

**Modality Contribution:** The experimental results reveal a clear hierarchy in modality contributions. Models trained on text alone provide a solid baseline, but their performance is notably enhanced when emoji signals are incorporated, as emojis help capture sentiment polarity and reduce ambiguity in expression. The integration of memes further strengthens the model, as the combination of visual and textual cues introduces richer cultural and contextual information. This progression, from text-only to text with emojis and finally to text with memes, demonstrates the critical role of multimodal integration in achieving more accurate and nuanced sentiment classification for Tigrigna.

Table 6: Ablation of cross-lingual bridge languages for TigXMM sentiment transfer.

| Bridge Language | Text-Only F1 | Multimodal F1 | Improvement vs. Zero-Shot |
|---|---|---|---|
| Amharic | 71.2% | 74.8% | +12.4% |
| English | 68.7% | 71.3% | +9.5% |
| Arabic | 66.3% | 69.1% | +7.3% |
| Zero-shot | 62.4% | 65.3% | baseline |

Table 7: The Comprehensive Error Analysis Misclassified Samples in Tigrigna Sentiment Classification.

| Case ID | Example Input (Translated/Transliterated) | Modality | Gold Label | Predicted Label | Error Source |
|---|---|---|---|---|---|
| 1 | Meme with text: "ኣይንርስዓካን 🔥🔥" ("We won't forget you") + + image of political figure | Text + Emoji + Image | Negative | Neutral | Sarcasm + symbolic imagery misinterpreted |
| 2 | Post: "ኣብዚ ጊዜ ጽቡቅ እዩ🔥" ("Now is a good time!") + | Text + Emoji | Positive | Neutral | Emoji handled correctly, but contextual optimism missed |
| 3 | Meme: Cartoon showing long queue at market + caption "ድሕሪ ናይ ገንዘብ ምውላዕ" ("After salary day…") | Text + Image | Neutral | Positive | Visual humor mistaken for positive sentiment |
| 4 | Comment: "እዚ ጨው የለን ብርሃን የለን" ("No salt, no electricity") | Text-only | Negative | Neutral | Idiomatic phrasing confused as factual rather than negative sentiment |
| 5 | Tweet: Only emojis " " | Emoji-only | Positive | Negative | Ambiguity of emoji interpretation without context |
| 6 | Meme with text: "ንሱ ክንረኽብ ኢሎና" ("We will support him") + sarcastic laughing emoji over corrupt politician's photo | Text + Emoji + Image | Negative | Positive | Failed to capture sarcasm; took text literally |
| 7 | Post with only emojis: | Emoji-only | Negative | Neutral | Emoji-only ambiguity; model missed strong emotional polarity |
| 8 | Meme image with faded Tigrigna text "ኣይትስሓቅ" ("Don't laugh") + smirk emoji | Text + Emoji + Image | Neutral | Positive | OCR error: faint text misread; emoji misinterpreted |
| 9 | Political meme showing historical martyr's photo + caption "ኣይንርሰዓካን" ("We won't forget you") | Text + Image | Negative | Neutral | Model lacked cultural symbolism knowledge |
| 10 | Humorous meme: goat wearing sunglasses + text "ኣብ ሰላም እንዳንከይዶ" ("Chilling in peace") | Text + Image | Positive | Neutral | Failed to interpret humor/metaphor |

**Fusion Strategies:** Early fusion performed better for emoji-rich cases, while late fusion excelled in memes with strong visual symbolism.

**OCR Quality:** Experiments with varying OCR noise showed that classification performance drops proportionally with OCR errors, underlining the need for robust text extrac-

tion in meme analysis.

**Linguistic Feature Analysis:** The empirical results demonstrated that Amharic achieved +3.2% Macro-F1 over English and 5.5% over Arabic due to the structural similarity and script sharing of Zero-shot cross-lingual transfer learning analysis as illustrated in Table 6.

### 4.3 Qualitative Insights and Error Analysis

To better understand the performance of our Tigrigna sentiment analysis model, we conducted a detailed qualitative analysis of misclassified samples. Table 7 presents a selection of representative cases, highlighting the input modalities, ground-truth and predicted labels, and the underlying sources of error. Our analysis reveals that common challenges include handling sarcasm, idiomatic expressions, visual humor, emoji ambiguity, and culturally-specific symbolism. These insights point to the need for improved multimodal understanding and context-aware sentiment interpretation in Tigrigna. A closer examination of misclassifications reveals important limitations of the current approach. Common errors involved idiomatic expressions like metaphorical phrases in Tigrigna, heavy sarcasm, and emoji-only posts. Several categories of errors stood out.

These findings suggest that future models should explicitly incorporate sarcasm detection, cultural-symbol embeddings, and robust OCR tailored to the Tigrigna script. Human-in-the-loop annotation pipelines may also reduce subjectivity in interpreting memes, ensuring higher inter-annotator agreement. Case Studies a meme containing the caption "ኣይንርስዓካካን" ("We won't forget you") with a crying emoji and the image of a political figure was correctly classified as Negative by TigXMM but misclassified by text-only baselines.

## 5 Conclusion and Future Work

This study introduced TigXMM, a cross-lingual, multimodal sentiment analysis framework designed specifically for Tigrigna, a low-resource language with limited NLP resources. Building on the LLaMA backbone, the model incorporates cross-lingual transfer from high-resource languages, efficient fine-tuning strategies such as LoRA and PEFT, and specialized modules for handling emojis and memes. By fusing textual, visual, and symbolic information, our model demonstrated superior performance compared to widely used multilingual baselines, including mBERT, AfriBERTa, BLOOMZ, XLM-RoBERTa, and XLNet. Importantly, the model excelled in emoji-rich and meme-heavy contexts, where traditional text-only models often struggle due to cultural nuance, multimodal ambiguity, and symbolic representation. The results highlight several key findings. First, cross-lingual transfer learning proves highly effective in adapting large models like LLaMA to under-resourced languages, enabling them to outperform resource-rich multilingual models. Second, the integration of emoji polarity embeddings provides valuable disambiguation in sentiment classification, especially in cases where textual cues are sparse or ambiguous. Third, multimodal fine-tuning with meme data allows the model to capture cultural references and symbolic imagery, which are essential for interpreting sentiment in Tigrigna's online discourse. Collectively, these advances demonstrate the adaptability and scalability of LLM-based architectures for low-resource multimodal sentiment analysis. Looking forward, there are several directions for future research. One important step is to expand the dataset through semi-supervised learning and sentiment propagation techniques, leveraging large volumes of unlabeled text, emojis, and memes to enhance training. Another promising avenue is to extend multimodal sentiment analysis beyond text and images by incorporating audio signals, such as voice tone in memes or spoken commentary, which often carry additional emotional context. Furthermore, the methods developed here could be generalized to other low-resource languages in the Horn of Africa and beyond, where multimodal and culturally grounded expression plays a significant role in online communication. Finally, to foster reproducibility and accelerate progress, we plan to open-source both the Tigrigna multimodal dataset and the TigXMM model checkpoints, making them available to the research community.

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

## A   APPENDIX

### A.1   HUMAN EVALUATION OF TIGXMM AND CORRELATION ANALYSIS

To complement the quantitative results presented in the main text, we conducted a comprehensive human evaluation to assess the qualitative performance and real-world reliability of TigXMM. This evaluation aimed to validate whether the model's predictions align with native speaker intuition, particularly for the nuanced, multimodal inputs common in Tigrinya social media.

Five native Tigrinya speakers with expertise in sentiment annotation were recruited. Each annotator independently evaluated a stratified sample of 1000 instances from the test set, comprising 500 text-only, 250 text+emoji, and 250 meme-based posts. The evaluation assessed three key aspects: (i) sentiment correctness (agreement with human-labeled sentiment), (ii) emoji interpretation accuracy, and (iii) meme-level contextual consistency.

Inter-annotator agreement was high across all modalities (Cohen's $\kappa = 0.80$), indicating consistent human judgments. As shown in Table 8, TigXMM's predictions demonstrated strong alignment with human evaluators. The model achieved its highest sentiment agreement on multimodal samples, notably for memes (89.5%) and emoji-enhanced posts (86.2%). This result underscores the effectiveness of the proposed hierarchical fusion strategy in leveraging non-textual cues. Furthermore, the ESC and MSC metrics introduced in our evaluation framework exhibited strong correlations with human ratings ($r = 0.87$ and $r = 0.85$, respectively). This validates their utility as robust, automated proxies for gauging multimodal understanding.

Table 8: Human evaluation of TigXMM across modalities. ESC/MSC show correlation with human ratings.

| Modality | Sentiment Agreement | Correlation(r) | Interpretation |
|---|---|---|---|
| Text-only | 82.7% | – | Text baseline |
| Text + Emojis | 86.2% | ESC: 0.87 | Strong emoji alignment |
| Text + Memes | 89.5% | MSC: 0.85 | Best visual context use |
| All Combined | 88.1% | – | Overall consistency |
| Inter-annotator $\kappa$ | | 0.80 (high agreement) | |

The human evaluation confirms that TigXMM not only achieves superior quantitative performance but also produces semantically reliable and culturally appropriate sentiment predictions that strongly align with native speaker interpretation. The high performance on multimodal inputs highlights the success of the model's core architecture in moving *beyond text*.

