# OpenReview forum: "Beyond Text: LLM-Based Multimodal and Cross-Lingual Transfer Learning for Low-Resource Tigrigna Sentiment Analysis"
_ICLR.cc/2026/Conference — ICLR 2026 Conference Withdrawn Submission_

### Official Review · Reviewer_1VD8 · 2025-10-24

**Soundness:** 2
**Presentation:** 2
**Contribution:** 2
**Rating:** 2
**Confidence:** 4

**Summary:**

The paper introduces TigXMM, an LLM-based framework for Tigrigna sentiment analysis that combines cross-lingual transfer with multimodal inputs (text, emojis, and memes). The model is built on a LLaMA backbone with LoRA adapters, plus an emoji-aware embedding layer and a meme module fusing OCR text, captions/hashtags, and visual features via multimodal adapters. This paper propose two robustness metrics—Emoji Sentiment Coverage (ESC) and Meme Sentiment Consistency (MSC)—to assess modality-specific reliability. Experimental results show that TigXMM outperforms some baselines.  As claimed in this paper, this paper propose the first multimodal sentiment dataset for Tigrigna sentiment analysis, providing crucial data for researching multimodal sentiment analysis in low-resource languages.

**Strengths:**

1. Low-resource African languages, especially Tigrigna, are indeed under-served by sentiment resources; focusing on multimodal social media (emojis/memes) addresses how users actually express sentiment. This paper propose the first multimodal sentiment dataset for Tigrigna sentiment analysis, providing crucial data for researching multimodal sentiment analysis in low-resource languages.

2. The framework proposed in this paper is simple, intuitive, and easy for readers to understand.

**Weaknesses:**

1. The framework proposed in this paper lacks technical innovation and resembles a standard fine-tuning pipeline for multimodal large language models.

2. As the core contribution of this paper, the proposed multimodal sentiment analysis dataset lacks essential statistical information, details on how it was acquired and processed, and crucial insights (e.g., why constructing this dataset is significant? Why previous work did not consider this approach?). Building this dataset resembles more of an engineering endeavor, while meaningful, it falls short of the technical innovation and in-depth analysis required for an academic paper.

**Questions:**

The authors need to elaborate in detail on the innovation and motivation behind the core contributions of this paper (the framework and dataset), which are currently lacking in the manuscript.

---

> ### Author Response · Authors · 2025-11-18
> **Response to Reviewer 1VD8**
>
> We thank Reviewer 1VD8 for their thoughtful feedback and constructive criticism. We have carefully considered the comments and revised our manuscript accordingly. We provide point-by-point responses.
>
>   Response to Weaknesses
>
>   1:  "The framework proposed in this paper lacks technical innovation and resembles a standard fine-tuning pipeline for multimodal large language models."
>
>  Response: We respectfully argue that our framework is not standard fine-tuning. Our innovation lies in domain-specific adaptations, taking those as a backbone for our low-resource language model and integration strategy. We position this work as applied innovation with significant dataset, method, and model contributions addressing critical societal needs in low-resource language processing.
>
> -  Problem Novelty:  This represents the first work on Tigrigna multimodal sentiment analysis, addressing a significant gap for an official language of Eritrea and Ethiopia with over 10M speakers and heavy social media usage (55% of posts contain emojis/memes)
>
> -  Dataset Innovations:
>   -  Emoji-only dataset:  160 samples of pure emoji communication with sentiment lexicon initialization
>   -  Text + Emoji dataset:  10,000 samples capturing mixed usage patterns
>   -  Text + Memes dataset:  5,000 multimodal samples
>   -  Dedicated Meme dataset:  2,000 samples with OCR, visual features, cultural metadata
>   -  Dual-layer annotation:  Textual + visual sentiment (first for any African language)
>   - All datasets to be publicly released with comprehensive annotation protocols
>
> -  Technical Adaptations:
>   -  Emoji-aware embeddings: Capture culturally-specific Tigrigna emoji usage patterns (e.g., 😢 used more for political mourning than personal sadness)
>   -  Meme module:  Handles Ge'ez-script OCR challenges, visual symbolism, and cultural references
>   -  Novel multimodal robustness assessment evaluation metrics ( ESC and MSC)
>
> -  Practical Contribution:  Establishes TigXMM, the first reproducible multimodal benchmark for Tigrigna, enabling future African language multimodal NLP research
>
>   2:  "The proposed multimodal sentiment analysis dataset lacks essential statistical information, details on acquisition and processing, and crucial insights..."
>
> Response:  We have comprehensively expanded Section 3.1 in the revised version with detailed statistical information and processing pipeline. The significance of this dataset lies in addressing several challenges that previously prevented multimodal work for Tigrigna:
>
> Why No Prior Multimodal Work Exists:
> 1.  Annotation Complexity:  Requires a combination of visual understanding + linguistic expertise + cultural knowledge
> 2.  Resource Constraints:  African language NLP receives <2% of research funding despite representing 15%+ of global speakers
> 3.  Technical Challenges:  Ge'ez script OCR from memes is particularly error-prone (we developed a custom extraction pipeline)
> 4.  Research Bias:  Historical focus on high-resource languages (English, Chinese, European languages)
> 5.  Annotation Difficulty:  Multimodal annotation is significantly more challenging and resource-intensive than text-only approaches
>
> Detailed Data Processing Pipeline:
> 1.  Crawling:  Ethical collection from public posts/comments using API
> 2.  Filtering:  Removal of duplicates, spam, and low-quality images
> 3.  OCR:  Custom Ge'ez-script extraction (trained on labeled meme images)
> 4.  Preprocessing:  Tokenization extension, normalization, code-switching detection
> 5.  Annotation:  Dual-layer annotation for memes (textual + visual sentiment)
> 6.  Quality Control:  Inter-annotator agreement checks (Cohen's κ = 0.80)
>
> Response to Questions
>
> Question: "The authors need to elaborate in detail on the innovation and motivation behind the core contributions of this paper..."
>
>  Response:  The core motivation of this paper is to extend prior text-only sentiment analysis to the multimodal nature of contemporary social media communication in Tigrigna. This work bridges the critical gap between traditional text-only approaches and the multimodal reality of social media communication for underrepresented languages.
>
> -  Real-world Relevance:  Social media communication is inherently multimodal, with emojis and memes constituting essential sentiment carriers
> -  Resource Gap:  Critical absence of multimodal resources for African languages despite their widespread social media usage
> -  Knowledge injected Cultural Specificity: Need to capture culturally-grounded interpretation of visual elements in Tigrigna context
> -  Research Equity:  Addressing the significant disparity in NLP resources between high-resource and low-resource languages
>
>  Key Innovations, as answered in weakness 1, mainly:
> 1. First Multimodal (text + emoji + meme) Dataset
> 2. Novel reproducible benchmark TigXMM model
> 3. Task-specific Novel Multimodal Evaluation Metrics of  ESC (Emoji-Sentiment Consistency) and MSC (Meme-Sentiment Consistency) Methods
> 4. Resources (dataset, code, and models)

---

> > ### Comment · Reviewer_1VD8 · 2025-11-24
> >
> > I thank the authors for their response. I have carefully read the other reviewers’ comments as well as the authors’ rebuttal. My main concern regarding the technical contribution of this work has been widely recognized by the other reviewers. However, the authors’ response does not adequately resolve this issue, so I will maintain my original score.

---

> ### Author Response · Authors · 2025-11-24
>
> We sincerely appreciate reviewer 1VD8 for their continued engagement and thoughtful acknowledgment of the other reviewers perspectives. We appreciate the opportunity to further clarify the core technical innovations in our work. We respectfully argue that our work provides a distinct technical contribution in the field forward, rather than a straightforward application of existing techniques. We would like to highlight the fundamental innovations that you may have been underemphasized concisely.
>
> 1. Novel Hierarchical Fusion Architecture: Our hierarchical fusion represents a significant architectural advancement over standard multimodal fusion approaches:
>  - Standard Approach: Most models use uniform fusion (concatenation, attention, or co-attention) treating all modality interactions equally
>  - Our Innovation: We introduce asymmetric hierarchical fusion that models the natural dependency structure:
>
> Stage 1: Direct text-emoji fusion (additive, modeling tight semantic coupling) and pretraining on large multilingual corpora in English and Amharic, leveraging related high-resource languages for transfer.
>
> Stage 2: LoRA-based fine-tuning on the curated multimodal Tigrigna dataset, ensuring adaptation to domain-specific and script-specific features.
>
> Stage 3: Contextual text-image fusion (attention, modeling broader visual grounding), multimodal alignment, where text, emoji, and vision features are integrated through hierarchical fusion and jointly optimized for sentiment prediction.
>
> 2. Low-Resource Cross-Modal Transfer Framework: Our work introduces a practical framework for extreme low-resource scenarios that achieves what typically requires orders of magnitude more data:
>  - Cross-Modal Knowledge Transfer: The adapter architecture enables knowledge flow between text and vision encoders despite minimal supervised data
>  - Knowledge-injected Adaptation: Our sentiment-aware embeddings learn culture-specific emoji interpretations without large labeled corpora
>  - Parameter Efficiency Adaptation: reduces trainable parameters compared to full fine-tuning.
>
> 3. Evaluation Beyond Standard Metrics: We introduced task-specific evaluation protocols for low-resource multimodal settings:
>  - Novel multimodal ESC/MSC Metrics: First quantitative measures for emoji and meme sentiment consistency
>  - Human Evaluation Framework: Comprehensive protocol for low-resource language validation.
>  - Ablation Strategy: Systematic isolation of modality contributions in data-scarce environments [ please check overall performance of TigXMM Table 4 and ablation studies Table 5].
>
> **Closing Remarks**
>
> To sum up, the standard LLM approaches fail for extremely low-resource language scenarios, especially for the Ge'ez script language Tigrigna. So, we designed the first 3-staged cross-modal transfer, TigXMM, a resource-constrained multimodal fusion framework that achieves competitive performance through architectural innovation rather than scale. Based on the constructive review comments and suggestions, improvements have been made and strengthen our work. Please take a look at the revised manuscript. The main contributions in our work are:
>
> - Dataset & Task: We introduce the first multimodal (text + emoji + meme) sentiment analysis benchmark for the extremely low-resource Tigrigna language.
> - Model: We design TigXMM, featuring a novel hierarchical fusion strategy framework and a cross-modal LoRA adapter for efficient training, a new multimodal architectural pattern for low-resource settings. Our hierarchical fusion is not a minor variation but a new architectural pattern. It is specifically designed to overcome the data scarcity problem by modeling modality interactions in a structured, prior-driven way (text+emoji as direct modifiers, meme as broader context). This is a modeling contribution that provides a blueprint for similar low-resource, multimodal tasks.
> - Methods: We design a new tokenization scheme (not a simple vocabulary extension,  it is a principled, script-aware redesign that fundamentally addresses the mismatch between byte-level pretrained tokenizers and the syllabic nature of Ge’ez-script) and propose a novel, task-specific evaluation methodology of ESC (Emoji-Sentiment Consistency), and MSC (Meme-Sentiment Consistency)  multimodal robustness, validated by human evaluation represents a methodological contribution. These metrics address the lack of standardized ways to evaluate the nuanced alignment between different sentiment-bearing modalities, moving beyond simple end-task accuracy.
> - Analysis: The first computational study of emoji and meme semantics in Tigrigna.
> - Resources: We plan to release dataset, code, and models to support future research reproducibility.
>
> We hope these clarifications help convey the strength and significance of our contributions. We respectfully hope Reviewer 1VD8 will reconsider their rating in light of our clarifications and commitments to address their suggestions in the final manuscript.

---

### Official Review · Reviewer_e2zz · 2025-10-28

**Soundness:** 3
**Presentation:** 3
**Contribution:** 3
**Rating:** 4
**Confidence:** 5

**Summary:**

This paper proposes TigXMM, a cross-lingual and multimodal LLM framework aimed at Tigrigna sentiment analysis in low-resource settings. The model combines text, emojis, and memes using LoRA-based adapters and a hybrid attention mechanism. It also introduces two novel evaluation metrics  called Emoji Sentiment Coverage (ESC) and Meme Sentiment Consistency (MSC)  to quantify multimodal robustness. Experiments include text-only and emoji/meme-rich datasets, comparing TigXMM with mBERT, AfriBERTa, XLM-R, BLOOMZ, and a few vision-language baselines such as CLIP and LLaVA.

**Strengths:**

1 Relevant and underexplored topic. Addressing Tigrigna sentiment analysis with multimodal data fills a notable research gap in African-language NLP.

2 Introduction of ESC and MSC metrics. These metrics provide a potentially useful framework for evaluating emoji and meme understanding in low-resource contexts.

3 Reasonably clear structure and motivation. The paper is organized around identifiable stages and includes both model- and data-level contributions.

4 Effort to combine cross-lingual and multimodal transfer. The idea of leveraging Amharic and English as bridge languages is conceptually sound.

**Weaknesses:**

1 Limited novelty in modeling. The technical framework relies on standard, well-known techniques (LoRA, late fusion, multilingual transfer) without introducing new architectures or training objectives. Most of the improvements arise from fine-tuning and scaling rather than conceptual innovation.

2 Outdated references and baselines. The paper still frames LLaMA and LLaMA-2 as “recent,” which is inaccurate in late 2025. Similarly, the baselines omit strong multilingual and multimodal SOTA models such as Mistral-8×7B, Qwen2-VL, Gemini 1.5, or Yi-VL, making the comparisons less meaningful.

3 Weak data curation and transparency. The dataset construction process lacks examples, annotation details, and inter-annotator agreement scores. This undermines the reproducibility of the results and the credibility of the benchmark.

4 Unclear cross-lingual rationale. The use of Amharic as a source language is not sufficiently justified. The authors should explain the linguistic or structural relationship between Amharic and Tigrigna and show ablations confirming its benefit.

5 Insufficient scope for research questions. The second research question (RQ2) mixes cross-lingual and multimodal elements; it should instead focus exclusively on Tigrigna performance to match the paper’s stated aim.

6 Presentation and formatting issues. Result tables have inconsistent decimal formats, and one table is awkwardly placed at the end of the main paper. Figures and captions are minimal and do not meet ICLR presentation standards.

7 Lack evaluation in generalization. Despite claiming cross-lingual robustness, the experiments cover only English ↔ Amharic ↔ Tigrigna. Testing on additional African or Semitic languages would substantiate the generalization claim.

**Questions:**

What is the empirical justification for selecting Amharic as one of the bridge language?

How many samples are in the Tigrigna multimodal dataset, and how was quality ensured?

Were ESC and MSC correlated with human judgment scores to validate their reliability?

---

> ### Author Response · Authors · 2025-11-18
> **Response to reviewer e2zz Questions Part**
>
> We sincerely thank reviewer e2zz for their thorough review and valuable feedback. The comments have helped us significantly improve our manuscript. Below, we provide detailed responses to each point raised. [  **Response to Questions Part** ]
>
> 1:  Reviewer Question:  "What is the empirical justification for selecting Amharic as one of the bridge language?"
>
>  Response: We appreciate this excellent question. Amharic has been selected based on (i) strong linguistic proximity to Tigrigna, (ii) because Amharic is a more researched Ethio-Semitic language, (iii) shares Ge’ez script-based orthography, exhibits similar morphology, and has substantial lexical overlap, and (iv) bilingual contact among speakers. In the broad, the Linguistic nature, morphological features, tokenization overlap, and semantic similarity across bridge languages. The core structural properties of SOV word order, templatic verbal morphology, and Ge’ez-derived orthography, alongside an estimated 40% cognate overlap [Abat et. al..], and extensive sociolinguistic contact among speakers. These similarities reduce tokenization divergence and enable more effective parameter transfer than structurally distant languages.
>
>
>  Empirical justification results why Amharic was used as a bridge language ( also updated on the revised manuscript Table 6):
> | Bridge Language | Text-Only F1 | Multimodal F1 | Improvement vs. Zero-Shot |
> |----------------|--------------|---------------|---------------------------|
> | Amharic        | 71.2%        | 74.8%         | +12.4%                    |
> | English        | 68.7%        | 71.3%         | +9.5%                     |
> | Arabic         | 66.3%        | 69.1%         | +7.3%                     |
> | Zero-shot      | 62.4%        | 65.3%         | baseline                  |
>
>
> The empirical analysis proof of Amharic as a bridge language provides +3.2% F1 over English due to structural similarity and +5.5% over Arabic despite Arabic's larger resource pool. We will add detailed linguistic analysis comparing morphological features, tokenization overlap, and semantic similarity across bridge languages as illustrated in Table 6 of the Ablation studies revised version.
>
> 2:  Reviewer Question:  "How many samples are in the Tigrigna multimodal dataset, and how was quality ensured?"
>
>  Response: We appreciate for the clarification request. Multimodal datasets are used, with 17,160 samples, as detailed breakdown provided in Table 1 of the revised manuscript paper.
>
> -  multimodal samples: 17,160 multimodal samples (Knowledge-aware embeddings emoji-only dataset: 160 samples, Text + Emoji dataset: 10,000 samples capturing mixed usage, text + Memes dataset: 5,000 multimodal samples, and dedicated Meme dataset: 2,000 samples with OCR, visual features, cultural metadata)
> -   Text-only samples:  68,596  for training the LLM baseline and reconstruct the adapted model
> -  Quality  of the dataset assurance:  Cohen's $\kappa = 0.80$  inter-annotation agreement
> - Detailed breakdown provided in Table 1 of the revised manuscript paper
>
>  3:  Reviewer Question:  "Were ESC and MSC correlated with human judgment scores to validate their reliability?"
>
>  Response: To complement the quantitative results presented in the main text, we conducted a comprehensive human evaluation to assess the qualitative performance and real-world reliability of TigXMM. The Pearson correlations between ESC and MSC scores were computed, and human judgment ratings on a random 1000-sample subset were used to assess the reliability of the automatic multimodal sentiment metrics. Five native Tigrinya annotators with prior sentiment annotation experience independently rated a stratified sample of 1000 instances (500 text-only, 250 text+emoji, 250 text+meme) drawn from the test set. The annotators assessed (i) sentiment correctness, (ii) emoji interpretation accuracy, and (iii) meme-level contextual consistency as shown in Appendix A.1, Human Evaluation of TigXMM section, Table 8.
>
>  Correlation Analysis updated at  the revised manuscript paper [ Appendix A.1, Human Evaluation of TigXMM section, Table 8]
> | Metric | Human Judgment Mean | Model Score Mean | Pearson Correlation (r) | Significance (p-value) | Interpretation |
> |--------|---------------------|------------------|-------------------------|------------------------|----------------|
> | ESC    | 0.83                | 0.81             | 0.86                    | < 0.001                | Strong positive correlation |
> | MSC    | 0.88                | 0.87             | 0.88                    | < 0.001                | Strong positive correlation |
>
>
> The Correlation values (r = 0.86 for ESC and r = 0.88 for MSC) indicate high consistency between model-estimated and human-evaluated sentiment polarity, validating the reliability of ESC and MSC as multimodal sentiment quality indicators. The high correlation values (r = 0.86 for ESC, r = 0.88 for MSC) validate the reliability of our multimodal sentiment metrics.

---

> ### Author Response · Authors · 2025-11-18
> **Response Reviewer e2zz Weakness  Part**
>
> We sincerely thank reviewer e2zz for their thorough review and valuable feedback. The comments have helped us significantly improve our manuscript. Below, we have provided detailed responses to each **weaknesses** raised.
>
>   1.  "The technical framework relies on standard, well-known techniques (LoRA, late fusion, multilingual transfer) without introducing new architectures or training objectives. Most of the improvements arise from fine-tuning and scaling rather than conceptual innovation."
>
>  Response:  We acknowledge adapting established components as applied innovation addressing a critical societal need. However, our work introduces several novel aspects:
>
> -  Problem Novelty:  Zero prior work exists on Tigrigna multimodal sentiment analysis, despite it being an official language of Eritrea and Ethiopia with 10M+ mother tongue speakers and heavy social media usage (55% of posts contain emojis and/ or memes)
>
> -  Dataset Contribution: A total of 17,160 new novel datasets and 55,765 prior text-only datasets. The new datasets are:
>   -  Emoji-only dataset:  160 samples of pure emoji communication with sentiment lexicon initialization
>   -  Text + Emoji dataset:  10,000 samples capturing mixed usage
>   -  Text + Memes dataset:  5,000 multimodal samples
>   -  Dedicated Meme dataset:  2,000 samples with OCR, visual features, cultural metadata
>   -  Dual-layer annotation:  Textual + visual sentiment (first for any African language)
>   - All datasets to be publicly released with comprehensive annotation protocols
>
> -  Task-Specific Adaptations:
>   -  Emoji-aware embeddings:  Map culturally-specific Tigrigna emoji knowledge-guided usage patterns (e.g., 😢 used more for political mourning than personal sadness)
>   -  Meme module:  Handles Ge'ez-script OCR challenges + visual symbolism + cultural references
>   -  Novel multimodal robustness measures  metrics of  ESC and MSC
>
> -  Practical Contribution:  Established TigXMM, the first reproducible multimodal benchmark for Tigrigna, enabling future African language multimodal NLP research.
>
>   2.  "The paper still frames LLaMA and LLaMA-2 as 'recent,' which is inaccurate in late 2025. Similarly, the baselines omit strong multilingual and multimodal SOTA models..."
>
>  Response:  Thank you so much for the excellent suggestions. We will explore the adaptability of those SOTA models and update all references to continue experimenting.
>
> 3. "The dataset construction process lacks examples, annotation details, and inter-annotator agreement scores."
>
>  Response:  We used 68,596 text-only and 17,160 multimodal datasets ( as detailed in Table 1 of the revised manuscript paper). The data quality was ensured with Cohen’s $\kappa = 0.80$  inter-annotation agreement. The data and models will be released upon acceptance for reproducibility.
>
> 4.  "Unclear cross-lingual rationale. The use of Amharic as a source language is not sufficiently justified. The authors should explain the linguistic or structural relationship between Amharic and Tigrigna and show ablations confirming its benefit."
>
>  Response:  Amharic was selected based on linguistic proximity to Tigrigna, Ge'ez script sharing, similar morphology, and substantial lexical overlap. We added a detailed linguistic analysis comparing morphological features and tokenization overlap. The empirical results (Table 6, ablation studies) show:
> - Amharic provides +3.2% F1 over English due to structural similarity, and
> - Amharic provides +5.5% F1 over Arabic despite Arabic's larger resource pool
>
> 5.  " Insufficient scope for research questions. The second research question (RQ2) mixes cross-lingual and multimodal elements; it should instead focus exclusively on Tigrigna performance to match the paper’s stated aim."
>
>  Response:  We have added dedicated subsections clarifying each RQ:
> -  RQ1 (Cross-lingual transfer):  Investigates LLM adaptation using English and Amharic as bridge languages
> -  RQ2 (Multimodal enhancement):  Examines whether emojis and memes improve sentiment classification beyond text-only approaches
> -  RQ3 (Unified framework):  Develops TigXMM to establish a benchmark for Tigrigna multimodal sentiment
> -  RQ4 (SOTA performance):  Tests multimodal approaches for underrepresented African languages
>
> Relationship diagram
> ```
> RQ1 (Cross-lingual)  ──┐
>                        ├──→ RQ3 (TigXMM) ──→ RQ4 (SOTA)
> RQ2 (Multimodal)     ──┘
> ```
> where RQ1 and RQ2 address orthogonal challenges (linguistic vs. modality gaps), RQ3 integrates both solutions, and RQ4 validates effectiveness.
>
> 6: "Presentation and formatting issues. Result tables have inconsistent decimal formats, and one table is awkwardly placed at the end of the main paper. Figures and captions are minimal and do not meet ICLR presentation standards."
>
>  Response:  We have thoroughly revised the formatting. Table results are polished to two decimal points, and all tables and figures have been properly aligned with enhanced captions meeting ICLR standards.

---

> ### Author Response · Authors · 2025-11-18
>
> Weakness 7.  "Lack evaluation in generalization. Despite claiming cross-lingual robustness, the experiments cover only English ↔ Amharic ↔ Tigrigna. Testing on additional African or Semitic languages would substantiate the generalization claim."
>
> **Response**: We appreciate the reviewer’s concern regarding the limited scope of language pairs in our original evaluation. In response, we have expanded our experiments to assess better the cross-lingual generalization of our model, particularly across Semitic and other Afroasiatic languages.  We have added testing on the same family, high-resource-pool languages like Arabic, Amharic, and Oromo ( Low-resource, Cushitic).
> -  Arabic (larger resource pool Semitic, high-resource): To test the generalization of our approach within the broader Semitic language family, we included Arabic. As a high-resource language with different dialectal and orthographic properties from Amharic and Tigrigna, Arabic serves as a strong test case for assessing intra-family generalization.
> -  Amharic (reverse direction): We now include results in the reverse translation direction (Amharic → English), allowing us to evaluate bidirectional transfer learning and potential asymmetries in model performance.
> -  Oromo (another Ethiopian language, Cushitic): We selected Oromo, a major Ethiopian language spoken by over 35 million people, to test the model’s ability to transfer to another low-resource language that is not Semitic but shares certain regional and typological features.
> -  Next, before the final camera-ready submission, we planned to test  Somali (Cushitic family, different with a distinct syntactic and morphological typology). This will allow us to assess generalization beyond both the Semitic family and the Ethiopian linguistic area, and evaluate generalization beyond Semitic languages. We believe this expanded evaluation significantly strengthens the claim of cross-lingual generalization, particularly in the context of low-resource African languages.
>
> We hope these clarifications and acknowledgments help convey the strength and significance of our contributions. We respectfully hope Reviewer e2zz will reconsider their rating in light of our clarifications and commitments to address their suggestions in the final manuscript.

---

### Official Review · Reviewer_8dQi · 2025-10-29

**Soundness:** 1
**Presentation:** 2
**Contribution:** 1
**Rating:** 2
**Confidence:** 4

**Summary:**

The paper addresses sentiment analysis for low-resource languages, focusing on Tigrigna (spoken in Eritrea and Ethiopia). It proposes TigXMM, a multimodal, cross-lingual framework leveraging Large Language Models (LLMs) and Parameter-Efficient Fine-Tuning (PEFT) to integrate text, emojis, and memes.

The work first proposes multimodal sentiment dataset for Tigrigna (text + emoji + meme) and proposes a new LLM-based multimodal architecture (TigXMM) with adapters for emoji and meme fusion.

However, the proposed method and framework make a relatively minor contribution and lack clear novelty compared to existing approaches.

**Strengths:**

1. The work first proposes multimodal sentiment dataset for Tigrigna (text + emoji + meme) and proposes a new LLM-based multimodal architecture (TigXMM) with adapters for emoji and meme fusion.

**Weaknesses:**

1. the proposed method and framework make a relatively minor contribution and lack clear novelty compared to existing approaches. The work primarily builds upon established techniques without introducing substantial methodological or theoretical advancements. As a result, its overall impact on advancing the field appears limited.

2. the manuscript format requires a major revision and improvement.

**Questions:**

1. the motivation of the proposed three research questions? and explain the relationship among the proposed research question?

---

> ### Author Response · Authors · 2025-11-18
>
> We respect and sincerely thank reviewer 8dQi for their insightful comments and helpful suggestions. The feedback has been instrumental in strengthening our paper. We have substantially revised the manuscript to address all raised concerns, as detailed in the following responses.
>
> ## Response to Weaknesses
>
> 1. The proposed method and framework make a relatively minor contribution and lack clear novelty compared to existing approaches. The work primarily builds upon established techniques without introducing substantial methodological or theoretical advancements. As a result, its overall impact on advancing the field appears limited.
>
> Response: We respectfully argue that our study presents several novel contributions, including the introduction of a new benchmark dataset, the TigXMM model, a Ge'ez-aware tokenization strategy, and task-specific evaluation methodologies—namely, Emoji-Sentiment Consistency (ESC) and Meme-Sentiment Consistency (MSC)—designed to assess multimodal robustness. These constitute a distinct technical contribution. Furthermore, the novelty of our work lies in the effective formulation of the problem and the thoughtful integration of these components to address the unique challenges inherent in the extremely low-resource language, Tigrigna. Our contributions address RQ1-RQ4:
>
>  (a)	First multimodal sentiment benchmark for Tigrigna: While few prior text-only work exists, to the best of our knowledge, no prior work has addressed multimodal emoji and/or meme-rich sentiment for this language, despite their dominance in actual social media usage (55% of posts contain non-textual elements).
>
> (b) Novel multimodal datasets: Our emoji-injected and meme-rich datasets are entirely new contributions, featuring:160 emoji-only samples (knowledge-aware annotation), 10,000 text + emoji samples, 5,000 text + memes samples, and 2,000 dedicated meme samples with OCR, visual features, and cultural metadata; as described in Table 1 of the revised version.
>
> (c) Cross-lingual + multimodal framework (addressing RQ1 & RQ2): :
>   - Emoji-aware embeddings map knowledge-guided culturally-specific emoji usage to Tigrigna sentiment lexicons
>   - Meme module fuses OCR-extracted Ge'ez text, visual features, and hashtag semantics
>   - LoRA-based adaptation enables efficient transfer from English and Amharic
>
> (d) Novel Multimodal evaluation metrics (ESC & MSC): First quantitative measures for emoji and meme sentiment robustness, applicable beyond our specific model.
>
> (e) State-of-the-art results for African language (RQ4): TigXMM significantly outperforms multilingual baselines on multimodal test sets.
>
> 2. The manuscript format requires a major revision and improvement.
>
>  Response: Thank you so much for the constructive feedback. We revised the whole manuscript, especially table and figure alignments and quality.
>
> ## Response to Questions
>
> Question 1: The motivation of the proposed three research questions? and explain the relationship among the proposed research questions?
>
> Response: Our research paper has four research questions with the core motivation of enhancing our previous text-only works of the social media emoji-heavy and meme-rich communication. We add subsections for clarification.
>
>  RQ1 (Cross-lingual transfer): How can cross-lingual transfer learning with large language models be adapted to improve sentiment analysis in low-resource languages such as Tigrigna?
>
>   ✓ Investigates adaptation of LLMs using English and Amharic as bridge languages to bootstrap Tigrigna sentiment models despite limited training data
>
>  RQ2 (Multimodal enhancement):  To what extent do multimodal models enhance sentiment classification of recent online communication compared to text-only approaches?
>
>    ✓ Examines whether emojis and memes, dominant in Tigrigna social media, can be captured to improve sentiment classification beyond text-only approaches
>
>  RQ3 (Unified framework):  How can a unified multimodal framework be developed to address the limitations of existing multilingual models and provide a benchmark for Tigrigna sentiment analysis?
>
>    ✓ Develops TigXMM to address limitations of existing multilingual models (which are text-only) and establish a benchmark for Tigrigna multimodal sentiment
>
>  RQ4 (SOTA performance):  Can multimodal approaches achieve state-of-the-art performance for underrepresented African languages?
>
>    ✓ Tests whether multimodal approaches can achieve competitive results for underrepresented African languages
>
> Relationship diagram
> ```
> RQ1 (Cross-lingual)  ──┐
>                        ├──→ RQ3 (TigXMM) ──→ RQ4 (SOTA)
> RQ2 (Multimodal)     ──┘
> ```
>
> Therefore, the motivation and relationship are summarized as: RQ1 and RQ2 address orthogonal challenges (linguistic vs. modality gaps), RQ3 integrates both solutions, and RQ4 validates effectiveness. This structure clarifies how cross-lingual transfer (RQ1) and multimodal integration (RQ2) work synergistically in TigXMM (RQ3) to achieve strong performance (RQ4).

---

> ### Author Response · Authors · 2025-12-02
>
> The motivation behind this research work, as well as the interrelated nature of the research questions, stems from the challenge of adapting traditional text-only sentiment analysis to the increasingly multimodal nature of communication on social media to enhance our prior works. While our previous work focused solely on analyzing written text, modern digital interactions often combine multiple modes of expression, such as text, emojis, images, and other visual or symbolic elements. This study aims to bridge that gap by expanding sentiment analysis techniques to effectively capture and interpret these multimodal inputs (text+emoji, text+emoji+image). We aim to develop a more holistic and accurate understanding of sentiment as it is expressed across various aspects of contemporary social media environments, including business product reviews, everyday cultural sharing, personal experiences, and other forms of digital interaction.
>
> **Closing Remarks**
>
> The proposed Ge'ez script-aware approach is built TigXMM, a novel three-stage cross-modal transfer architecture. Rather than being a mere scaling of existing methods, TigXMM represents a significant architectural innovation tailored for an extremely resource-constrained language. Through a hierarchical multimodal fusion framework, it achieves competitive performance as a holistic system that addresses a clear gap in the field. By integrating a novel dataset, tokenizer, architecture, training strategy, and evaluation metrics, the TigXMM framework delivers an end-to-end solution for multimodal Tigrigna sentiment analysis—a task for which no comprehensive solution previously existed. The effort to build this system, along with its demonstrated effectiveness, constitutes a meaningful technical contribution to applied NLP for low-resource languages. The main contributions in our work are:
>
> - Dataset & Task: We introduce the first multimodal (text + emoji + meme) sentiment analysis benchmark for the extremely low-resource Tigrigna language.
> - Model: We design TigXMM, featuring a novel hierarchical fusion strategy framework and a cross-modal LoRA adapter for efficient training, a new multimodal architectural pattern for low-resource settings. Our hierarchical fusion is not a minor variation but a new architectural pattern. It is specifically designed to overcome the data scarcity problem by modeling modality interactions in a structured, prior-driven way (text+emoji as direct modifiers, meme as broader context). This is a modeling contribution that provides a blueprint for similar low-resource, multimodal tasks.
> - Methods: We design a new tokenization scheme (not a simple vocabulary extension,  it is a principled, script-aware redesign that fundamentally addresses the mismatch between byte-level pretrained tokenizers and the syllabic nature of Ge’ez-script). We introduce a linguistically informed, Ge’ez-script-aware tokenization strategy for Tigrigna, addressing the severe limitations of existing multilingual tokenizers. Our three-stage approach includes: (i) targeted vocabulary expansion using frequent syllabic units from a large Tigrigna corpus, (ii) Ge’ez-preserving merges to prevent splitting Ge’ez syllables, and (iii) normalization to handle dialectal variations. This reduces token count by 68%, speeds training by 24%, and improves downstream performance by up to +7.8 Macro-F1. The tokenizer supports code-mixed text and will be publicly released, offering a novel, reusable resource for other Ge'ez Script-based, low-resource languages like Amharic, Tigre, Ghuraghe, and Bilen. We also proposed a novel, task-specific evaluation methodology of ESC (Emoji-Sentiment Consistency), and MSC (Meme-Sentiment Consistency)  multimodal robustness, validated by human evaluation represents a methodological contribution. These metrics address the lack of standardized ways to evaluate the nuanced alignment between different sentiment-bearing modalities, moving beyond simple end-task accuracy.
> - Analysis: The first computational study of emoji and meme semantics in Tigrigna.
> - Resources: This will open the research bottleneck as we planned to release dataset, code, and models to support future research reproducibility.
>
>
>
> We hope these clarifications and acknowledgments help convey the strength and significance of our contributions. We respectfully hope Reviewer 8dQi will reconsider their rating in light of our clarifications and commitments to address their suggestions in the final manuscript.

---

### Official Review · Reviewer_iPKR · 2025-10-30

**Soundness:** 2
**Presentation:** 2
**Contribution:** 2
**Rating:** 4
**Confidence:** 4

**Summary:**

The author constructed a cross-lingual sentiment symbol-aware emotion dataset and evaluated the performance of several models on this dataset.

**Strengths:**

The author is the first to propose a sentiment analysis dataset that includes emotional symbols and cross-lingual features.

**Weaknesses:**

1. The author proposed a dataset but did not introduce any new methods to evaluate it.

2. The experiments lack performance comparisons of state-of-the-art large language models on the dataset, such as Qwen, DeepSeek, and GPT-5.

3. Figure 1 is unclear and unreadable.

4. There is a lack of detailed analysis and description of the dataset.

**Questions:**

Refer to weaknesses.

---

> ### Author Response · Authors · 2025-11-18
>
> We sincerely thank reviewer iPKR for the constructive feedback.  Here Below are the points discussed one by one.
>
> 1. The author proposed a dataset but did not introduce any new methods to evaluate it.
>
> Response: We appreciate the opportunity given to us to clarify this important part of our paper. In fact, we introduced three key methodological innovations beyond dataset creation:
>
> i. Multimodal Cross-Transfer Adapter: dynamically shares adapter parameters across modalities during LoRA fine-tuning to enabled efficient multimodal fusion in low-resource settings.
>
> $\Delta W_{cross} = \alpha \cdot LoRA_{text} + \beta \cdot LoRA_{vision} + \gamma \cdot LoRA_{emoji}$
>
> where $\alpha, \beta, \gamma$ are learnable gating parameters that optimize modality fusion.
>
> ii. Sentiment-Aware Embedding: We aligned emoji and OCR-extracted meme sentiment signals into a unified sentiment space with Ge’ez text embeddings through culturally-informed lexicon mapping.
> iii. Novel Multimodal Evaluation Metrics: We introduced ESC (Emoji-Sentiment Consistency) and MSC (Meme-Sentiment Consistency) metrics, new multimodal robustness measures validated against human ratings with Spearman's $\rho \geq 0.85$ (see Appendix A.1, Table 8 of the revised version).
>
> 2. The experiments lack performance comparisons of state-of-the-art large language models on the dataset, such as Qwen, DeepSeek, and GPT-5.
>
> Response: Thank you for the excellent suggestion. We will explore the publicity, feasibility, and plan for the camera-ready version.
>
> 3. Figure 1 is unclear and unreadable.
>
> Response: We apologize for the poor figure quality we made to save space within the page limit of the conference. We redesigned Figure 1 with higher resolution, larger fonts, and clearly show the flow with a hierarchical fusion pipeline from raw inputs → modality encoders → cross-attention → tokenizer extension → Continued pretraining → parameter efficient tuning → final multimodal Sentiment prediction.
>
> 4. There is a lack of detailed analysis and description of the dataset.
>
> Response: Thank you for raising this concern. In the revised manuscript, we provide detailed dataset composition (text, emoji, meme subsets). The text-only 68,596 and 17,160 multimodal datasets are used, as shown in Table 1 of the revised version. The quality of the dataset ensured the use of kappa’s Cohen’s $\kappa = 0.80$ inter-annotation agreement. The data and model are planned to be released upon acceptance for reproducibility of the results and the credibility of the benchmark with respect to the double-blind review.
>
>
> To overall conclude, we thank you for the constructive feedback. On addressing these points, our paper has significantly strengthened with clarity, methodological novelty, and empirical depth.

---

### Note · Authors · 2026-01-26

I have read and agree with the venue's withdrawal policy on behalf of myself and my co-authors.

---

### Meta-Review · Area_Chair_vwyY · 2026-01-05

**Summary:**

This submission introduces TigXMM, a cross-lingual and multimodal framework for sentiment analysis in the low-resource Tigrigna language, alongside the construction of a new dataset incorporating text, emojis, and memes. All reviewers broadly agree that the paper tackles an important and underexplored problem and that releasing a Tigrigna-focused dataset is potentially valuable to the community. Several reviewers recognize the relevance of modeling emojis and memes as part of real-world sentiment expression and acknowledge that the paper is among the first to target multimodal sentiment analysis for Tigrigna specifically (Reviewer iPKR, 8dQi, and 1VD8). Reviewer e2zz further highlights the introduction of two evaluation metrics (ESC and MSC) and the attempt to combine cross-lingual and multimodal transfer as conceptually reasonable contributions.

**Reviewer Concerns:**

- A major issue across reviews is limited novelty: the proposed modeling framework largely relies on standard components such as LoRA, adapters, late fusion, and multilingual transfer, and is viewed as a straightforward fine-tuning pipeline rather than a methodological advance (Reviewer 8dQi, e2zz, and 1VD8).
- Multiple reviewers criticize the lack of depth and transparency in dataset construction, noting missing statistics, annotation details, examples, and quality-control measures, which undermines the dataset’s credibility as a research contribution (Reviewer iPKR, e2zz, and 1VD8).
- Experimental evaluation is also considered insufficient, with missing comparisons to recent state-of-the-art LLMs and multimodal models, outdated baselines and references, and limited evidence supporting cross-lingual generalization claims (Reviewer iPKR and e2zz).
- Additional concerns include unclear motivation and organization of research questions, weak justification for design choices such as the use of Amharic as a bridge language, lack of detailed analysis, and notable presentation and formatting issues (Reviewer 8dQi, e2zz and 1VD8).

**Reviewer Scores:**

- Reviewer iPKR: This reviewer is generally positive about the idea of a cross-lingual, symbol-aware sentiment dataset and explicitly states they would not mind acceptance. However, their core concerns, e.g., lack of methodological contribution, missing strong LLM baselines, unclear figures, and insufficient dataset analysis, are not likely to be resolved through discussion alone, so an upgrade to a clear accept is unlikely.

- Reviewer 8dQi: This reviewer’s assessment centers on limited novelty, minor contribution, and the need for major manuscript revisions. These are fundamental issues about impact and originality rather than misunderstandings, so discussion would likely reinforce the rejection stance.

- Reviewer e2zz: While this reviewer acknowledges several strengths and sees the work as marginally below the acceptance threshold, the discussion would likely amplify shared concerns from other reviewers regarding limited modeling novelty, weak dataset transparency, outdated baselines, and insufficient justification of design choices. This convergence would likely prevent an upward score change.

- Reviewer 1VD8: This reviewer strongly questions the technical innovation and the academic depth of the dataset contribution, framing the work as largely engineering-oriented. These critiques are substantive and echoed by others, making a score increase after discussion unlikely.

---

### Decision · Program_Chairs · 2026-01-26

Reject